# Transcriptional Pausing and Activation at Exons-1 and -2, Respectively, Mediate the *MGMT* Gene Expression in Human Glioblastoma Cells

**DOI:** 10.3390/genes12060888

**Published:** 2021-06-08

**Authors:** Mohammed A. Ibrahim Al-Obaide, Kalkunte S. Srivenugopal

**Affiliations:** Department of Pharmaceutical Sciences, Jerry H. Hodge School of Pharmacy, Texas Tech University Health Sciences Center, Amarillo, TX 79106, USA; m.alobaide1950@gmail.com

**Keywords:** *MGMT*, DNA repair, promoter methylation, alkylating agents, brain tumors, transcriptional pausing, nuclear transcription assays

## Abstract

**Background:** The therapeutically important DNA repair gene O^6^-methylguanine DNA methyltransferase (*MGMT*) is silenced by promoter methylation in human brain cancers. The co-players/regulators associated with this process and the subsequent progression of *MGMT* gene transcription beyond the non-coding exon 1 are unknown. As a follow-up to our recent finding of a predicted second promoter mapped proximal to the exon 2 [*Int. J. Mol. Sci.*
**2021**, *22*(5), 2492], we addressed its significance in *MGMT* transcription. **Methods:** RT-PCR, RT q-PCR, and nuclear run-on transcription assays were performed to compare and contrast the transcription rates of exon 1 and exon 2 of the *MGMT* gene in glioblastoma cells. **Results:** Bioinformatic characterization of the predicted *MGMT* exon 2 promoter showed several consensus TATA box and INR motifs and the absence of CpG islands in contrast to the established TATA-less, CpG-rich, and GAF-bindable exon 1 promoter. RT-PCR showed very weak *MGMT*-E1 expression in *MGMT*-proficient SF188 and T98G GBM cells, compared to active transcription of *MGMT*-E2. In the *MGMT*-deficient SNB-19 cells, the expression of both exons remained weak. The RT q-PCR revealed that *MGMT*-E2 and *MGMT*-E5 expression was about 80- to 175-fold higher than that of E1 in SF188 and T98G cells. Nuclear run-on transcription assays using bromo-uridine immunocapture followed by RT q-PCR confirmed the exceptionally lower and higher transcription rates for MGMT-E1 and *MGMT*-E2, respectively. **Conclusions**: The results provide the first evidence for transcriptional pausing at the promoter 1- and non-coding exon 1 junction of the human *MGMT* gene and its activation/elongation through the protein-coding exons 2 through 5, possibly mediated by a second promoter. The findings offer novel insight into the regulation of *MGMT* transcription in glioma and other cancer types.

## 1. Introduction

O^6^-Methylguanine-DNA-methyltransferase (*MGMT*) is a simple, low molecular weight (~23 kDa), antimutagenic DNA repair protein expressed in varying amounts in normal tissues, but generally, at elevated levels in human malignancies including brain cancers [1,2]. *MGMT* is unique in that it can remove the mutagenic alkyl groups bound to the O6-position of guanine in a single-step stoichiometric reaction, thereby restoring the normal G-C base pairing, and effectively preventing the mispairing of O6-alkylguanines with thymine [3]. Because *MGMT* repairs the mutagenic and cytotoxic O6-alkylguanine adducts generated by the clinically used anticancer alkylating agents, it has emerged as a central and rational target for overcoming tumor resistance to alkylating agents [4,5]; this is particularly true for brain cancers because the hydrophobic monofunctional and bifunctional alkylating agents that cross the blood–brain barrier and generate O6-methyl/alkyl guanines in DNA remain the drugs of choice for this tumor type [6]. Both the antimutagenic and drug resistance properties conferred by *MGMT* derive from its ability to transfer the alkyl groups to an active-site cysteine (Cys145) in a self-inactivating suicidal reaction mechanism [7].

The *MGMT* gene, located on chromosome 10q26, occupies a region of 303,780 nucleotides and contains 5 exons and 4 introns [8]. Exon 1 and a large part of exon 5 are non-coding and the translated protein has its origin in exon 2 [8]. A 69 kb Intron-1 separates the exons 1 and 2. A minimal GC-rich promoter of 1.15 kb proximal to exon 1 was characterized in 1991 [9]. This promoter lacks TATA and CAAT box motifs but is notable for having a 777-bp CpG island with 97 CpG sites and an enhancer-binding site, and sites for the SP-1 and AP-1 transcription factors including a glucocorticoid response element [9,10]. It is now well established *MGMT* expression is epigenetically silenced through promoter methylation in a large subset of brain tumors [11], and *MGMT* has emerged as one of the best-studied candidates for its susceptibility to cytosine-phosphate-guanine (CpG) island (CGI) for DNA methylation in glioma and other cancers [12,13]. Many studies have shown a correlation between *MGMT* protein expression in gliomas and the methylation level of the promoter region, which is usually about 30 to 60% [14,15]. To be more specific, the *MGMT* promoter methylation levels are 20–30% for grade I gliomas, 60–80% for grade II, 40–50% for grade III, and 20–45% for grade IV tumors [16]. Studies in glioma cell lines comparing the *MGMT* expression and the extent of its promoter methylation have also been performed [17,18]; these investigations have identified two methylated areas in the CpG island: a highly methylated region upstream of the transcription initiation site, including a minimal promoter, and a highly methylated region downstream of exon 1 [19,20]. Because *MGMT* promoter methylation and the resulting decrease in *MGMT* expression in brain tumors are associated with a better response to chemotherapy, greater overall survival, and longer time to disease progression [21,22], currently, molecular assays are routinely employed in glioma specimens for therapeutic design and prognostic measures [23,24]. Methylation of CpGs located at the first noncoding exon and enhancer seems more critical for loss of *MGMT* expression; therefore, these sequences are probed in clinical assays [23].

The initiation of *MGMT* transcription is controlled by the *MGMT*-P1 promoter, NCBI-Nucleotide ID: X61657.1, mapped at the untranslated *MGMT* exon 1 [9]. Recently, we provided the first evidence for the presence of *MGMT* alternative promoters that overlap the *MGMT*-E1 promoter and motifs for p53, NRF1/2, CTCF, and the Myc-MAD-MAX genetic switch in the regulatory region [8]. Further, our search using the Transcriptional Regulatory Element Database (TRED; ref. [24]) showed the presence of a second predicted promoter, TRED-5071, referred to as the *MGMT*-E2 promoter henceforth. This sequence is located at the 5′-side of *MGMT*-E2 approximately 70 kilobases downstream of the *MGMT*-E1 promoter [8]. No information is available on the initiation and elongation of the *MGMT* transcripts in human cells and whether these events are altered in glioma to promote an active transcription or its attenuation. How exactly the promoter methylation intersects with these processes is also unclear. Therefore, extending our work, the present study investigated the role and significance of the second promoter in *MGMT* transcription in human glioblastoma cells.

## 2. Materials and Methods

### 2.1. Genomic Databases

Public databases NCBI-Nucleotide, Ensembl, UCSC, PrESSTo/FANTOM5, TRED, and EPD were searched for genomic criteria and map locations of *MGMT* promoters [8]. The UCSC genome browser Blast tool was used to retrieve and analyze sequences in the forward or reverse strands, and to recover the identified sequences from earlier versions to the updated GRCh38/hg38 version.

### 2.2. Transcription Factor Binding Sites (TFBSs) and CpG Islands

An in silico search was conducted to identify transcription factor DNA binding sites (TFBSs) involved in the initiation of transcription. The following motifs were included in the analysis: TATA-8 (TATAWA, TATAWAWR) and TATA-532, INR (YYANWYY), CCAAT and its inverted sequence TAACC, BRE (SSRCGCC), DPE (RGWCGTG), MYC/MAX/MAD genetic switch, and the GAGA factor (GAF) motifs The bioinformatic search was conducted using JASPAR, TRED, tfbss.org tools, and Bio-Base. The following parameter sets were used to identify CpG islands in the nucleotide sequence of the *MGMT* promoter: Observed/Expected ratio > 0.60, Percent C + Percent G > 55.00, Length > 200 bp.

### 2.3. Glioblastoma Cell Culture

Three human cell lines were used in this study. A pediatric glioma SF188 (Grade IV) cell line was purchased from the Department of Neurosurgery, Univ. of California, San Francisco. T98G (Grade IV) and SNB19 (highly invasive glioblastoma) cell lines were obtained from The American Type Culture Collection (Manassas, VA, USA). All cancer cell lines were grown in Dulbecco’s modified Eagle medium (DMEM) supplemented with 10% fetal bovine serum in a humidified 5% CO_2_ atmosphere at 37 °C.

### 2.4. Western Blotting (WB)

Standard Western blot procedures were followed [25]. Briefly, cell-free extracts of the three cell lines were prepared by sonication in the presence of protease inhibitors; equal protein amounts were combined with the SDS sample buffer followed by boiling for 1 min. The samples were then electrophoresed on 15% SDS-polyacrylamide gels, transferred to Immobilon-P membranes (Millipore Co.) followed by blocking in 2% BSA in PBS. The blots were incubated with monoclonal antibodies to MGMT protein, washed, exposed to anti-mouse IgG-HRP conjugate. Positive bands were visualized by enhanced chemiluminescence (ECL, GE Life Sciences). The protein bands were quantitated by densitometry using a VersaDoc 5000 imaging system and ImageJ software.

### 2.5. RNA Extraction, Conventional RT-PCR, and RT-qPCR

RNA was extracted from 70–80% confluent GBM cells using RNAzol reagent (GeneCopoea, Rockville, MD, USA). cDNAs were prepared from 0.5 µg of total RNA using specific primer sets for *MGMT*-exons 1, 2, and 5. The primer sets for the *MGMT*-E1 promoter were F: GCGCTCTCTTGCTTTTCTCA, R: GACACTCACCAAGTCGCAAA. The primers for the *MGMT*-E2 promoter were F: TGGAGCTGTCTGGTTGTGAG, R: TGGAAAACATGCCGTTATCA. The primers for extension of *MGMT*-exon 5 were F: CCGTGAAGGAATGGCTTCTG, R: TAGCTCCCGCTCCCTTG. Normalization was performed using the hypoxanthine-guanine phosphoribosyl transferase 1 (*HPRT1)* as the reference gene [26] using the primers F: TGAGGATTTGGAAAGGGTGT, R: GAGCACACAGAGGGCTACAA. The QIAGEN one-step RT-PCR kit was used for conventional RT-PCR according to standard protocols supplied by the manufacturer. The amplified cDNA fragments and DNA marker ladder of 100 bp (QIAGEN) were separated using 1.5% agarose gel and visualized by ethidium bromide staining. The cDNA image documentation and expression measurements were performed by densitometry using the VersaDoc Imaging System and ImageJ software. RT q-PCR reactions were performed in triplicate using the Taq Universal SYBR Green One-Step Kit (BIO-RAD) and quantified by the BIO-RAD iCylcer iQ system software. Normalization and relative expression analysis for the *MGMT* target gene was carried out using *HPRT1* as a reference gene [26] according to the Livak 2^−∆∆CT^ method, which was used to analyze relative gene expression data of real-time quantitative PCR [27].

### 2.6. Nuclear Run-On (NRO) Transcription Assays

Nuclear run-on transcription assays [28] were performed to investigate the transcriptional pausing in the *MGMT* gene and quantitate the relative rates of transcription initiated from the promoters proximal to *MGMT* exon 1 and *MGMT* exon 2, respectively. For these experiments, the non-isotopic bromouridine immunocapture nuclear run-on followed by the RT-qPCR method [29] was followed to achieve higher sensitivity and accuracy for capturing nascently labeled transcripts and determining transcription rates. Briefly, nuclei from SF-188 GBM cells (12 × 10^6^) growing in their logarithmic growth phase were isolated by gentle lysis in 10 mM Tris-HCl (pH 8.0) containing 0.5% NP-40, 0.5 mM MgCl_2,_ 1 mM dithiothreitol, 5% glycerol and low-speed centrifugation. Freshly isolated nuclei (5 × 10^6^/assay) were suspended in ice-cold assay buffer (10 mM Tris-HCl (pH 8.0), 2 mM dithiothreitol, 2.5 mM MgCl_2_, 150 mM KCl, 1 mM each of the nucleotides, ATP, GTP, CTP, 0.5 mM Bromo-UTP and 0.5 mM UTP. The reactions were incubated at 30 °C for 30 min. Following DNase and protease treatments, the total RNA was extracted using RNAzol and the nascent transcripts were purified by binding to a BrdU monoclonal antibody (Santa Cruz Biotechnology, Santa Cruz, CA, USA) and protein-G Dynabeads. The RNA eluted therefrom was converted to cDNA using random primers and a high-capacity cDNA Reverse Transcription Kit (catalog number: 4368814, Life Technologies, Carlsbad, CA, USA) [29]. The cDNA was quantitated by NanoDrop spectrophotometry. Next, qPCR reactions were set up with an equal amount of cDNA input (at 1X and 2X) and the primers listed below to quantitate the transcripts initiated at the *MGMT* promoters (1 and 2) and the corresponding exons as well. The sequences of the primer set for *MGMT*-E1 promoter, F: CGGGCCATTTGGCAAACTAA, R: CCCTTCGGCCGGTACAA. The primer sequences for *MGMT*-E1, F: CCCTAGAACGCTTTGCG, R: ACCCAGACACTCACCAA. MGMT-E2 promoter set, F: AAACGAGAAGAATCGGGATACAG, R: CCATTACCTAGAGCAGCCAAC. *MGMT*-E2 set, F: CAGCCTCTTACCTATACACTTTGTC, R: GGTGCGTTTCATTTCACAATCC. Normalization was performed using HPRT1 as a reference gene [26]. The HPRT1 F: CATTTACCACTTCTAGGCCCC, R: TCAGTCCATAACAAGCACCC. The real-time PCR reactions were performed in the iTaq universal SYBR Green reaction mix on a Bio-Rad CFX96 Real-Time System under the following thermal cycling conditions: polymerase activation and DNA denaturation at 95 °C for 3 min followed by 60 cycles consisting of denaturation at 95 °C for 15 secs, annealing and extension at 63 °C. Normalization and relative expression analysis for the *MGMT* target sequence were carried out using *HPRT1* as a reference gene according to Schmittgen and Livak’s 2^−∆∆CT^ method [27].

### 2.7. Statistical Analysis

The statistical analyses were performed using Excel software. T-tests were used for the analysis of two independent samples, whereas one-way ANOVA was used to analyze the differences among group means of more than two samples. All statistical computations were calculated using GraphPad Prism 7. A significant difference was assessed at *p* < 0.05.

## 3. Results

### 3.1. Genomic Context of MGMT Locus, MGMT Promoters, and Enhancer Sequences in the Vicinity

The *MGMT* gene locus occupies a region of 303,780 bps in the forward (plus) strand of the long arm of chromosome 10 at 10q26.3 and genomic coordinates (GRCh38/hg38): chr10: 129,467,184–129,770,983. The *MGMT-*mRNA (NM_002412) space of 1372 bp is comprised of five exons and one *MGMT* promoter X61657.1, whereas other databases showed five more overlapping promoter sequences that were combined by us to form a 2019 bp sequence called the *MGMT*-E1 promoter, mapped at 5′ of the untranslated *MGMT*-E1 [8]. In silico analysis of the revised *MGMT*-E1 promoter revealed the presence of a genetic switch MYC/MAX/MAD, and transcriptional pausing factor GAF motifs. This study analyzed the molecular features of the unexplored but predicted *MGMT*-E2 promoter (TRED-5071) located at 5′ of *MGMT*-E2 (Figure 1A). Since the gene enhancers and promoters often cooperate and function similarly to regulate gene expression [30], we also performed a search for enhancer sequences in the *MGMT* gene region using the VISTA Enhancer Browser (https://enhancer.lbl.gov, accessed on 14 May 2021.), a website listing experimentally validated enhancers. Enhancers are cis-acting regulatory sequences that function independently of orientation and present at various distances from their target promoters and act to increase gene transcription [31]. Interestingly, our search found four enhancers mapped in *MGMT* introns 2 and 3 and an additional four enhancers mapped in the genomic space between *MGMT* and MKI67 loci (Figure 1A, Table 1). Validating our efforts is the recent demonstration that a 3 kb enhancer located between the promoters of Ki67 (MKI67, a marker of cell proliferation) and *MGMT* genes serves to increase the *MGMT* expression significantly and confer temozolomide resistance [32]. The influence of the other eight enhancers (many of them near the Ki67/MKI67 gene) (Table 1) on *MGMT* transcription is unknown and requires further studies.

None of the enhancers above overlapped with the predicted *MGMT*-E2 promoter. The *MGMT*-E2 promoter showed several TATA and INR (initiator element) motifs spread all over the 1 kb sequence and at the upstream and downstream of the transcription start site (TSS) (Figure 1B). An ATF1 binding site found in the core promoters was also present. Unlike the well-characterized exon 1 promoter, the predicted *MGMT*-E2 promoter was distinct in lacking the CpG islands, GAF motifs, and other negative regulatory sequences. However, the E2 promoter was remarkable in having motifs routinely found in core gene promoters, thus providing a clue for its involvement in the *MGMT* transcription process. Accordingly, the *MGMT*-E2 promoter is not like *MGMT*-P1, which is prone to silencing by DNA methylation, and bears a distinct transcriptional pausing signature [8]. The sequence of the predicted *MGMT*-E2 promoter is shown in Figure 2. We propose that the *MGMT*-E2 promoter has a vital function in the transcriptional elongation and supports a robust synthesis of the *MGMT* gene transcript. Experimental evidence for this hypothesis has been provided in the sections to follow in human glioblastoma cell lines.

### 3.2. Expression of MGMT Promoters Proximal to Exon 1 and Exon 2 in MGMT-Proficient and -Deficient GBM Cell Lines

The human *MGMT* gene is rather unusual, with DNA methylation elements clustered in the promoter 1 and a non-coding exon controlling the expression of a transcript, with its first codon 67 kb away in exon 2. The molecular events regulating the initiation and elongation of the *MGMT* transcript are not known. In the context of our findings of a promoter-like sequence 5′ of exon 2, we performed a series of experiments to investigate the relative expression of exon 1 and exon 2 in three glioblastoma cell lines. Two MGMT-proficient and TMZ resistant (SF188 and T98G) and an *MGMT*-deficient/TMZ sensitive SNB19 cell line were selected for these studies. SNB19 cells have no detectable *MGMT* expression due to promoter methylation [33,34]. Western blotting for MGMT was performed as an initial step in these cell lines. The results confirmed SF188 and T98G cells to be MGMT-proficient, with the former having approximately five-fold more MGMT protein than T98G; SNB19 cells, as expected, did not show detectable levels of MGMT protein (Figure 3).

Next, we performed exon specific expression for *MGMT* for the first time using both conventional and real-time PCR procedures. While a whole-gene expression analysis provides overall information on transcription, measuring the activity of each exon can shed light on the nature of transcriptional initiation, elongation, and pause events [35]. Specific primers were designed to amplify 170–200 bp fragments in *MGMT*’s exon 1 and exon 2 in the three GBM cell lines; the DNA products obtained after routine RT-PCR and their relative expression are shown in Figure 4A,B, respectively. All cell lines SF188, T98G, and SNB19 generated strong intense bands for the cDNA amplicons of *MGMT*-E2 compared to the feeble bands for *MGMT*-E1 (Figure 3A). The SF188 cells showed higher expression of amplified cDNA fragments for both *MGMT*-E1 and *MGMT*-E2 relative to the T98G and SNB19 cells (Figure 4B). The RNA extracted from SNB19 did yield positive DNA bands for both MGMT-E1 and MGMT-E2, albeit at lower levels, indicating the *MGMT* mRNA could be produced, at least marginally in this cell line. One-way ANOVA showed a consistent and significantly increased expression of *MGMT*-E2 over *MGMT*-E1 in all cell lines (Figure 4B). Intriguingly, no clear correlation was evident between the MGMT protein and exon 1/2 expression levels, suggesting the operation of post-transcriptional processes.

### 3.3. RT q-PCR Analysis of MGMT-E1, MGMT-E2, and MGMT-E5 Expression

Further confirmation of the critical role played by the unexplored *MGMT*-E2 promoter and its comparison with the established *MGMT*-E1 promoter were obtained by quantitative real-time RT-PCR. Again, the cell lines SF188, T98G, and SNB19 were used. The analysis was based on the comparison of C_T_ (cycle threshold) value measurements for the transcription of three *MGMT* exons, *MGMT*-E1, *MGMT*-E2, and *MGMT*-E5. The C_T_ values are inversely proportional to the amount of target nucleic acid in the PCR reaction sample, i.e., the lower the C_T_ level indicates, the greater the target DNA amount in the PCR reaction sample. The calculated 2^−∆∆^C_T_ values [27] were used to verify the expression differences of the *MGMT* exons 1 and 2 relative to the expression of the calibrator reference gene, HPRT1. The data in Figure 5 and Figure 6 show that the relative normalized expression values for *MGMT*-E1 and *MGMT*-E2 promoters that reflect their original content in the whole transcript population in the three GBM cell lines. The results unambiguously reveal the poor and lower expression of *MGMT*-E1 compared to the strong and robust expression of *MGMT*-E2 in SF188 and T98G cells. The data also confirmed an exceptionally weak expression of the *MGMT*-E1 and *MGMT*-E2 in SNB19 cells, explaining the observed lack of *MGMT* mRNA and protein in these cells [33,34]. The evaluation of the average number of amplification cycles required to observe significant expression of the transcripts in four independent experiments in T98G cells showed a 65- to 70-fold lower expression of exon 1 compared to exon 2 (Figure 5D). In SF-188 cells, this difference was much greater with up to 90-fold greater expression of exon 2 relative to exon 1; additionally, exon 5, which is 230 kb away, was transcribed at rates similar to exon 2 (Figure 6). The expression differences between the *MGMT* E-1 compared with that of E2 and E5 in all three GBM cells remained significant at *p* < 0.05. Initiation of transcripts is always considered a rate-limiting step, and the evidence provided was strongly suggestive of transcriptional pausing at *MGMT*’s exon 1 and a rapid elongation along with exons 2 through 5 to generate a complete unprocessed transcript.

### 3.4. Quantitation of MGMT-E1 and MGMT-E2 Transcription Rates by Nuclear Run-On (NRO) Transcription Assays

The in vitro run-on transcription in nuclei is an established procedure to assess the direct activation of transcription. In this method, cells are lysed, and nuclei are prepared, halting the transcription by RNA polymerase. The transcripts already initiated in the nuclei are elongated in vitro up to 500 nucleotides, and the amount of labeled, nascent RNA at a given region corresponds to the abundance of RNA polymerase at the target site and provides accurate data on the rates of transcription [28,36]. Run-on transcription of the *MGMT*’s exon 1 and exon 2 was performed by incubating freshly isolated nuclei from SF-188 cells in the presence of ribonucleotides including Bromo-UTP. The nascent transcripts generated in these reactions were selectively purified on anti-BrdU affinity matrices, reverse transcribed to their cDNA, and the sequences corresponding to *MGMT*’s E-1 and E-2 were amplified by quantitative RTq-PCR. The primers for real-time PCR were carefully designed and included parts of the promoters and exons [29] to amplify the selected sequences. The locations of the designed primers marked by two-sided arrows for upstream and downstream *MGMT* exon 1 and exon 2 amplicons are shown in Figure 7. We referred to upstream or downstream exon 1 and exon 2 as *MGMT*-UE1 (P1), *MGMT*-DE1 (E1), *MGMT*-UE2 (P2) and *MGMT*-DE2 (E2), respectively.

The P1, E1, P2, and E2 transcripts in NRO bromouridine-labeled nascent RNA derived from nuclei of SF188 cells were measured by RT qPCR. The NRO itself and subsequent real-time PCR reactions were repeated twice, and similar results were obtained. We used the HPRT1 gene as the control and relative quantitation method, 2^−∆∆CT^ [27] as described in the previous section, to analyze the data. The quantitation shown in Figure 8A,B reveals that the transcription rate across exon 2 was at least 10 times higher than that of exon 1, which also reflects the enriched RNA polymerase content and rest of the transcription apparatus around exon 2. This direct evidence for efficient transcription from exon 2 and onwards is consistent and agrees well with the results of the RT-PCR and RT q-PCR presented, respectively, in Figure 4A,B, Figure 5A–D, and Figure 6, all pointing to a pause at the initiation step.

## 4. Discussion

Human *MGMT* remains a critical DNA repair gene at the crossroads of carcinogenesis, malignant progression, and anticancer therapies, particularly for brain cancers. We investigated the molecular aspects of *MGMT* transcription in human GBM cells and provided the first evidence for a tight transcriptional initiation, pausing, and a rapid elongation. An earlier bioinformatic and experimental work from our laboratory [8], demonstrating new regulatory sequences, transcription motifs, and downregulation of *MGMT* expression by CTCF, laid the foundation for the current study. Much information has established the core promoter extending into a non-coding exon of *MGMT* first described by Harris et al. [9] to be the hub of transcription initiation for the DNA repair protein. It is well-equipped to direct an accurate initiation of transcription by the RNA polymerase II in the absence of a TATA-box but having defined sites for the transcription factors (SP1, AP1, glucocorticoid response element), an enhancer, and silencers in the form of long stretches of CpG islands that are susceptible to methylation [37]. Research in our laboratory has located several other regulatory elements in the *MGMT* promoter, namely, the estrogen receptor element (ERE), antioxidant response element (ARE) which binds the transcription factors, NRF1 and NRF2, sites for the binding of GAGA pausing factor (GAF), a MYC/MAX/MAD genetic switch, CTCF and p53-binding consensus motifs in intron 1 [8,38]. Though the human *MGMT* gene has been considered a non-inducible constitutionally active gene [39], the above features bestow a tight and adequate regulatory role for the established promoter in controlling the *MGMT* expression. However, the mechanics of how this tight initiation is integrated with a productive elongation of *MGMT* mRNA and which genetic elements determine the elongation efficiency are yet unclear.

In this context, our bioinformatic search using in silico analysis of the TRED database [24] revealed the presence of a novel unexplored promoter proximal to *MGMT*’s exon 2 [8], and part of this study was aimed at validating its involvement in *MGMT* expression. The in silico analysis showed that the *MGMT*-E2 promoter is characterized by the absence of CGI, motifs for GAF factor, and MYC/MAX/MAD element, which are all associated with silencing of gene expression and transcriptional pausing [8]. On the other hand, the *MGM*T-E2 promoter showed abundant binding sites for INR (initiator element), ATF1 (activating transcription factor 1), BRE (the TFIIB recognition element), and a large number of TATA box motifs (Figure 2) that are regulatory sequences associated with a core promoter involved in active transcription [40]. We provided experimental data that confirmed the *MGMT*-E2 promoter is highly expressed compared to the feebly expressed *MGMT*-E1 promoter in SF188 and T98G, two known MGMT-proficient GBM cell lines. Further, we showed by RT q-PCR that *MGMT*-E2 and *MGMT*-E5 amplicons are expressed at comparable rates, which is an indication of the critical role played by the *MGMT* E2 promoter in *MGMT* expression and the formation of full-length *MGMT* mRNA transcripts. More definitive evidence for an exceptionally low transcription rate of *MGMT*-E1 contrasting with a highly efficient faster rate for *MGMT*-E2 was obtained by nuclear transcription assays, which elongate the pre-initiated transcripts for short stretches and provide accurate information on RNA polymerase occupancy at a given sequence.

Initiation by RNA Pol II has long been known to be a rate-limiting step in transcription. Our findings of a low rate of synthesis at *MGMT*’s exon 1 followed by a rapid and almost constant elongation burst through exon 5 clearly points to transcriptional pausing at the promoter–exon 1 interspace. In this process, the transcription machinery pauses downstream of the TSS before beginning a productive elongation. About 35% of protein-coding genes possess this type of regulation. The pausing process is a key component of human gene expression and has been suggested to be a checkpoint in early transcription where signals can be integrated for rapid and synchronous gene activation [41]. The human *c-myc* gene is a prototype example of transcriptional pausing, where an elongation block occurs within the first exon and has been studied extensively by nuclear run-on transcription analyses [42,43]. In many ways, the exon structure, the associated promoters, and transcriptional blockade we describe for human *MGMT* are very similar to that found in the *c-myc* gene in mammalian cells. For example, the c-myc gene harbors a first non-coding exon where the transcription freeze occurs [42,43]; moreover, an additional promoter in *c-myc* where productive initiation takes place has been described [44]. While exquisite pausing for *c-myc* which is inducible by growth factors, scores of signals, and cell cycle is understandable, the significance of such phenomena for the *MGMT* gene is not clear.

The regulatory elements and motifs present in *MGMT*’s promoter (P1) do provide a good explanation of why it is a preferred location for transcriptional pausing. There are three copies of GAGA-associated factor (GAF) motifs (GAGAG and GAGAGA) in the *MGMT* promoter, of which two are located upstream and one just downstream of the TSS [8]. The primary function of GAF is to facilitate transcription pausing [45,46], and mutations in the GAGA sequences lead to a loss of Pol II pausing [47]. The *MGMT* promoter also hosts a motif for a MYC/MAX/MAD repression network [8] where MAD/MAX complexes can antagonize the transcription [48]. Furthermore, the role of CGIs in acquiring methylation and silencing the *MGMT* gene is well established [12,13]. In contrast, the *MGMT* promoter 2 proximal to exon 2 does not have negative regulatory elements and, therefore, is well-positioned to support continued mRNA elongation.

We would like to stress that more detailed studies focusing on shorter segments of the *MGMT* promoter 1-exon 1 are required to pinpoint the exact location of RNA polymerase pausing and its subsequent release. We did not perform experiments to evaluate the direct role of the E2 promoter; however, the data reported indeed suggest an important function for this sequence. Further studies are also required to define the nature, role, and fine interplay of transcription regulatory proteins at the second promoter using ChIP and Cap Analysis of Gene Expression (CAGE) [49] analyses. Noteworthy also is that the SNB GBM cells which are *MGMT-*deficient did express small amounts of exon 1 and exon 2, perhaps indicating the instability of *MGMT* mRNA in these cells (Figure 4). Finally, the presence of a large number of intragenic enhancers in the *MGMT* genomic region and those between the *MGMT* and MKI loci (Table 1) adds to the complexity of *MGMT* gene expression in a given tissue or malignancy. A deeper understanding of *MGMT* transcriptional mechanisms underlying the transcriptional pause and activation, as highlighted here, may provide new avenues to curtail *MGMT* expression in gliomas.

## Figures and Tables

**Figure 1 genes-12-00888-f001:**
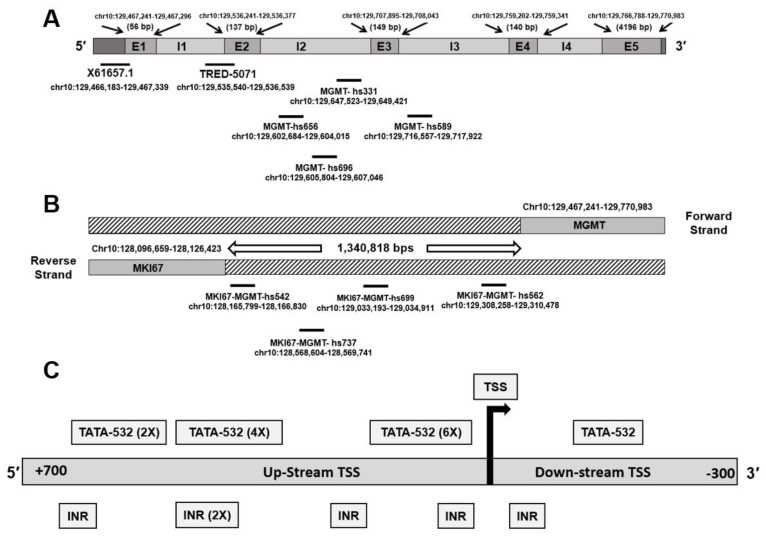
(**A**) *MGMT*-E1 promoter (X61657.1) mapped at 5′ of exon 1 and the predicted *MGMT*-E2 promoter (TRED-5071) mapped at 5′ of *MGMT*’s exon 2 are shown. The positions of four enhancers mapped in the *MGMT* intron 2 and intron 3 regions are also represented. (**B**) The four intergenic enhancers mapped to the genomic space between *MGMT* and MKI67 loci are shown. (**C**) Locations of TATA-532 motifs and transcription factor binding motifs in the predicted *MGMT*-E2 promoter. The relative positions of the promoters, enhancers, and transcription pausing motifs revealed from our study are also shown in the Graphical Abstract.

**Figure 2 genes-12-00888-f002:**
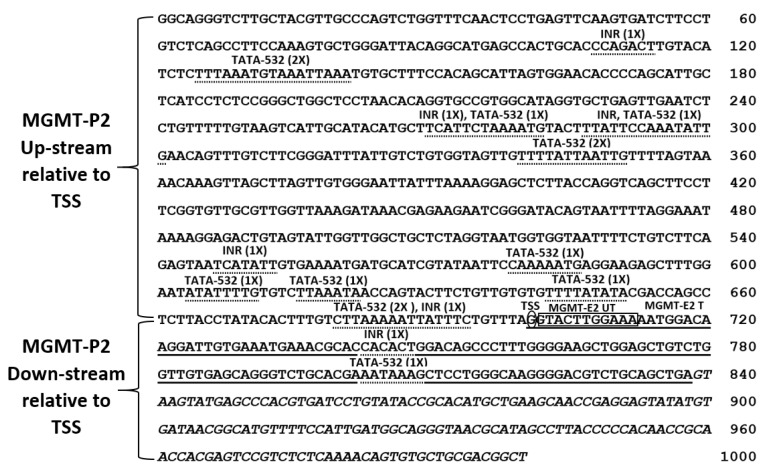
The 1000 nucleotide sequence of the predicted *MGMT*-E2 promoter (TRED-5071). The solid line represents the MGMT-E2 sequence, ‘T’ marks the *MGMT*-E2 translated sequence, ‘UT’ marks the untranslated sequence of *MGMT*-E2, italic letters mark the *MGMT* intron-1, TSS is the predicted transcription starting site, and dotted lines mark the TATA-532 and INR motifs.

**Figure 3 genes-12-00888-f003:**
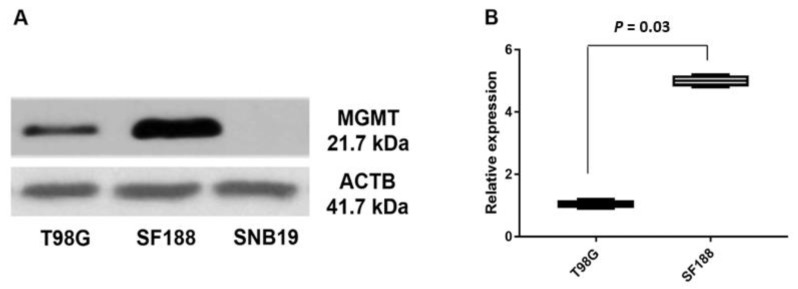
(**A**) Western blot of MGMT in T98G, SNB19, and SF188 GBM cells. (**B**) Relative protein expression values in the T98G and SF188 cell lines (*p* < 0.05).

**Figure 4 genes-12-00888-f004:**
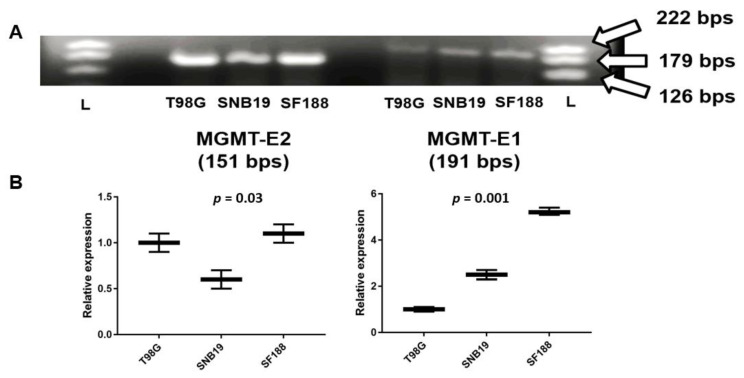
RT-PCR analyses of *MGMT* E-1 and *MGMT* E-2 expression in T98G, SNB19, and SF188 cells. (**A**) Amplicons visualized after agarose gel electrophoresis and EtBr staining are shown. (**B**) One-way ANOVA analyses of *MGMT*-E2 and *MGMT*-E1 relative expression (*p* < 0.05).

**Figure 5 genes-12-00888-f005:**
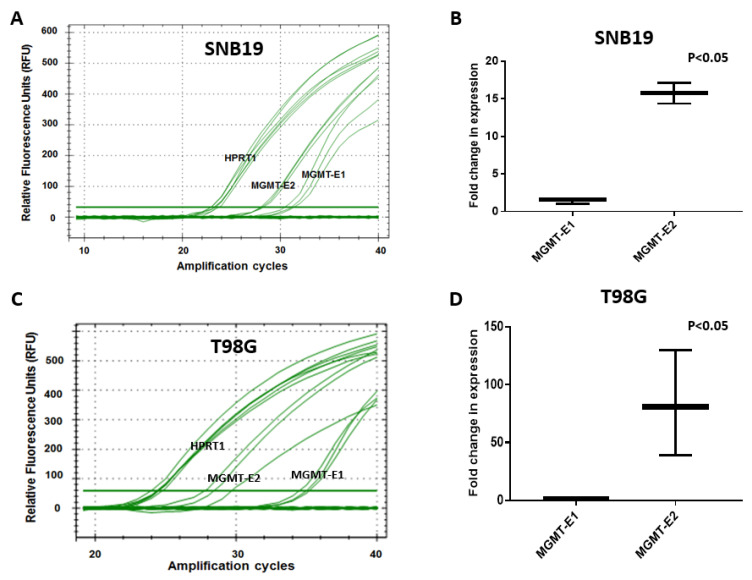
RT q-PCR expression measurements of *MGMT*-E1 and E2 in human GBM cell lines T98G and SNB19. Normalization and relative expression analysis for *MGMT* target sequences was carried out using HPRT1 as a reference gene [26]. (**A**,**B**) The relative fluorescence unit (RFU) values and fold change in expression of *MGMT*-E1 versus *MGMT*-E2 relative to the expression of reference gene HPRT1 in SNB-19 and T98G cells are shown. (**C**,**D**) Quantitative representation of the RFU values and fold change in expression for the two cell lines are shown.

**Figure 6 genes-12-00888-f006:**
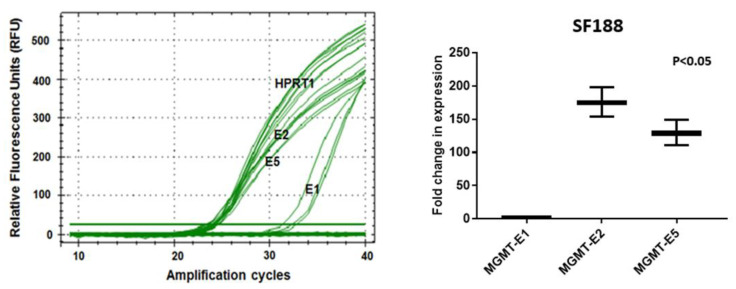
RT q-PCR expression measurements of RFU values for *MGMT*-E1, E2, and E5 in SF188 cells. Normalization and fold change in expression were performed relative to the *HPRT1* according to Schmittgen and Livak’s 2^−ΔΔC^_T_ method [27].

**Figure 7 genes-12-00888-f007:**
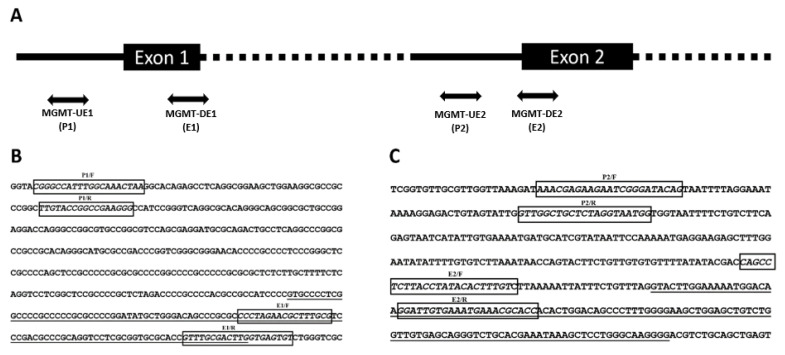
The two-sided arrows indicate the designed primers upstream or downstream of exon 1 and exon 2: *MGMT*-UE1 (P1, promoter 1), *MGMT*-DE1 (E1), *MGMT*-UE2 (P2, promoter 2), and *MGMT*-DE2 (E2) were used for amplification of nascent transcripts obtained from the marked regions by nuclear run-on (NRO) and subsequent RT q-PCR assays. (**A**) Representation of *MGMT*-E1 and *MGMT*-E2 genomic regions. (**B**) Sequences of the forward and reverse primer set, P1/F and P1/R for the upstream *MGMT*-E1 and E1/F and E1/R for downstream *MGMT*-E1. (**C**) Sequences of the forward and reverse primer set, P2/F, and P2/R for upstream *MGMT*-E2 and E2/F and E2/R for downstream *MGMT*-DE2 (*MGMT*-DE2). Forward and reverse primer sequences are shown in italics. *MGMT*-E1 and *MGMT*-E2 sequences are underlined.

**Figure 8 genes-12-00888-f008:**
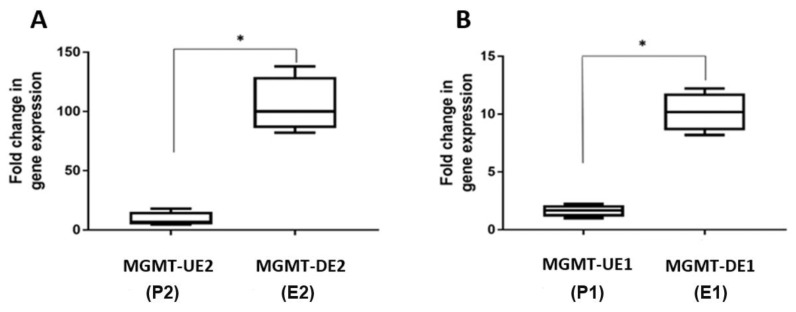
Nuclear run-on (NRO) transcriptional activities of SF188 *MGMT*-E1 and *MGMT*-E2 measured by RT qPCR. Normalization and relative expression analysis for MGMT target sequences were performed as described previously [27]. (**A**) Fold change in the transcription rates of *MGMT*-UE2 (P2) and *MGMT*-DE2 (E2). (**B**) Fold change in the transcription rates of *MGMT*-UE1 (P1) and *MGMT*-DE1 (P2) fold change in gene expression, * = *p* < 0.05.

**Table 1 genes-12-00888-t001:** The regulatory sequences including two promoters and nine enhancers located within the *MGMT* genomic space, and between the *MGMT*-*MKI-67* genes.

Type	ID	Bracketing Genes	Coordinates (hg38)	References
Promoter	*MGMT*-E1 promoter (X61657.1)	*MGMT* (5′ UTR)	chr10:129,466,183–129,467,339	Harris et al. 1991 [9]
promoter	*MGMT*-E2 promoter (TRED-5071)	*MGMT*	chr10:129,535,540–129,536,539	TRED-5071, [8] and this study
Enhancer	*MGMT*-hs331	*MGMT* (intragenic)	chr10:129,647,523–129,649,421	VISTA Enhancer Browser https://enhancer.lbl.gov, accessed on 14 May 2021.
Enhancer	MKI67-*MGMT*-hs542	MKI67-*MGMT*	chr10:128,165,799–128,166,830	
Enhancer	MKI67-*MGMT*- hs562	MKI67-*MGMT*	chr10:129,308,258–129,310,478	
Enhancer	*MGMT*- hs589	*MGMT* (intragenic)	chr10:129,716,557–129,717,922	
Enhancer	*MGMT*-hs656	*MGMT* (intragenic)	chr10:129,602,684–129,604,015	
Enhancer	*MGMT*-hs696	*MGMT* (intragenic)	chr10:129,605,804–129,607,046	
Enhancer	*MKI67*-*MGMT*-hs699	*MKI67*-*MGMT*	chr10:129,033,193–129,034,911	
Enhancer	*MKI67*-*MGMT*- hs737	*MKI67*-*MGMT*	chr10:128,568,604–128,569,741	
Enhancer	*MKI67*-*MGMT*	MKI167- *MGMT*	Chr10:130,704,894– 130,708,206	Chen et al. 2018 [32]

## Data Availability

Details of data from all databases and bioinformatic information presented in this study will be shared. Further, the cell lines and materials used here will be available on request.

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
