# Peer review of "Transcriptional Pausing and Activation at Exons-1 and -2, Respectively, Mediate the MGMT Gene Expression in Human Glioblastoma Cells"

_genes, 2021, doi:10.3390/genes12060888_

Round 1

Reviewer 1 Report

Dear authors,

First of all, I’d like to give a great congratulation to them for nice and successful study. I think that the topic and idea is novel enough to attract much interest to the readers. Also, their study was well designed, and methods were also reasonable and scientific. Their study gives an answer why certain glioblastoma patients have poor outcome even they have methylated promotor of MGMT. They did a good job. I think it can be frosting on the cake that they can illustrate several clinical cases (two or three glioblastoma patients cases) which validate their data.

Good luck.

Author Response

Dear authors, First of all, I’d like to give a great congratulation to them for nice and successful study. I think that the topic and idea is novel enough to attract much interest to the readers. Also, their study was well designed, and methods were also reasonable and scientific. Their study gives an answer why certain glioblastoma patients have poor outcome even they have methylated promotor of MGMT. They did a good job. I think it can be frosting on the cake that they can illustrate several clinical cases (two or three glioblastoma patients cases) which validate their data. Good luck

Response:  We appreciate the reviewer’s excellent and insightful comments.  The suggestion to demonstrate MGMT transcriptional pausing in GBM specimens is valid and will be pursued in a separate study. Thank you.

Reviewer 2 Report

In the work, „Transcriptional Pausing and Activation at Exons-1 and -2 Respectively Mediate the MGMT Gene Expression in Human Glioblastoma Cells“, the authors Mohammed A. Ibrahim Al-Obaide and Kalkunte S. Srivenugopal present a very interesting series of studies elicitating new insight into the regulation of expression of the MGMT gene which is of substantial importance concerning the malignant role and treatment resistance of glioblastoma cells.

The work includes both in silico and in vitro work leading to new insight in the understanding of these mechanisms, possible opening the chance for new treatment opportunities.

Only some minor aspects might be commented on.

Three cell lines - two MGMT proficient, one MGMT deficient (or promoter silenced) - cell lines were used; it might be discussed whether this is sufficient to cover the whole lot of different glioblastoma cases which are observed, known to be very heterogeneous in their genetic makeup. In particular, the role of the MGMT-E2 promoter, which has been described only recently, might be different in different types of cell lines.

Positioning of the three cell lines in figs. 3 and 4 is different, making comparison a little difficult.

Text formatting is inconsistent within in the text and should be corrected.

Elsewise, the work appears very well-written and important.

Author Response

In the work, „Transcriptional Pausing and Activation at Exons-1 and -2 Respectively Mediate the MGMT Gene Expression in Human Glioblastoma Cells“, the authors Mohammed A. Ibrahim Al-Obaide and Kalkunte S. Srivenugopal present a very interesting series of studies elicitating new insight into the regulation of expression of the MGMT gene which is of substantial importance concerning the malignant role and treatment resistance of glioblastoma cells.

The work includes both in silico and in vitro work leading to new insight in the understanding of these mechanisms, possible opening the chance for new treatment opportunities.

Response:  We are thankful and welcome  the reviewer's encouraging remarks.

 Criticism: 

Only some minor aspects might be commented on.

Three cell lines - two MGMT proficient, one MGMT deficient (or promoter silenced) - cell lines were used; it might be discussed whether this is sufficient to cover the whole lot of different glioblastoma cases which are observed, known to be very heterogeneous in their genetic makeup. In particular, the role of the MGMT-E2 promoter, which has been described only recently, might be different in different types of cell lines.

Response:  

We agree that a large number of cell lines would have been better and variations may occur between glioma cell lines and in patients.  However, the expression differences between exon-1 and exon-2 were so large (>180- fold in SF188) and highly reproducible.  We believe these differences will persist in a larger study.

Criticism: Positioning of the three cell lines in figs. 3 and 4 is different, making comparison a little difficult.

Response:  We agree. In response, we have brought the Figures 3 and 4  next to each other for better comparison.

Criticism: Text formatting is inconsistent within in the text and should be corrected.

Response: We have edited the manuscript and formatting errors have been minimized.

Elsewise, the work appears very well-written and important.

Respone: Thank you Sir. We appreciate the comment.